# Multiple criteria decision analysis for therapeutic innovations in a hemophilia care center: A pilot study of the organizational impact of innovation in hemophilia care management

Karen Beny[1,2]* , Amélie Dubromel[2] , Benjamin du Sartz de Vigneulles[1] , Valérie Gay[3], Florence Carrouel[1], Claude Negrier[4], Claude Dussart[1,2]

1 Health Systemic Process (P2S), Research Unit 4129, University Claude Bernard Lyon1, University of Lyon, Lyon, France, 2 Central Pharmacy, Lyon Public Hospices, Lyon, France, 3 Hemophilia Care Center, Métropole Savoie Hospital, Chambery, France, 4 Reference Center on Hemophilia and other Constitutional Hemorrhagic Diseases, Groupement Hospitalier Est, Lyon Public Hospices, Lyon, France

☯ These authors contributed equally to this work.
* karen.beny@chu-lyon.fr

**Data Availability Statement:** All relevant data are within the paper.

## Abstract

### Background

Several innovative drugs liable to lead to changes in healthcare organization are or soon will be available for the management of hemophilia. Analyzing their implementation can shed further light on healthcare decision-making, to anticipate changes and risk of breakdown in the patient's care pathway.

### Methods

Multiple criteria decision analysis (MCDA), based on ISPOR recommendations, was used to assess the organizational impact of innovation in hemophilia care management. The MCDA process designed for this specific context involved ten French experts in hemophilia care management (physicians, nurses, pharmacist, physiotherapist and psychologist) in the hemophilia care center of Chambéry, in the Rhône-Alpes Region of France. This pilot study involved seven steps: (i) defining the decision problem; (ii) selecting and structuring criteria; (iii) assessing the relative weight of each criterion with software-assisted simulation based on pairwise comparisons of different organizational change scenarios; (iv) measuring the performance of the selected innovations; (v) scoring alternatives; (vi) calculating aggregate scores; (vii) discussion. The endpoint was to determine the expected overall organizational impact on a 0–100 scale.

### Results

Seven organizational criteria were selected. "Acceptability for patient/caregiver/association" was the most heavily weighted. Factor VIII by subcutaneous route obtained the highest

**Funding:** The author(s) received no specific funding for this work.

**Competing interests:** The authors have declared that no competing interests exist.

aggregate score: i.e., low impact on care organization (88.8 out of 100). The innovation with strongest organizational impact was gene therapy (27.3 out of 100).

## Conclusion

This approach provided a useful support for discussion, integrating organizational aspects in the treatment decision-making process, at healthcare team level. The study needs repeating in a few years' time and in other hemophilia centers.

## Introduction

Until the turn of the 21st century, hemophilia treatment basically consisted in repeated intravenous replacement of the deficient coagulation factor [1]. For some years now, new therapies have been emerging with different, non-replacement action mechanisms, less burdensome administration route (subcutaneous), longer action duration and, in the case of gene therapy, a curative aim [2–5]. Some are real technological innovations liable to disrupt care organization and/or patients' care pathways. Management of hemophilia, which is a rare condition, is highly structured and governed by guidelines for treatment and care organization. Patients are managed in specialized structures ensuring global multidisciplinary care. In France, the structures are hospital-based and the clotting factor concentrates are available only in hospital pharmacies. The patient's pathway is thus highly hospital-centered; but this is expected to change, as, in France, new treatments become available in community pharmacies, closer to the patient. These less burdensome treatments and accompanying clinical improvements are liable to move care away from specialized structures and to involve new actors in patient management [6]. To maintain optimal secure care, these organizational impacts need to be studied to accompany the changes in pathway.

Introducing an innovation impacts both the organization governing it and its environment [7]. The organizational aspect is still not fully taken into account, but is emerging as one of the key aspects of the decision-making process, alongside the clinical and medico-economic aspects. Assessment of therapeutic innovations (healthcare products, acts, care organization) does not at present take account of all aspects of healthcare organization, and this can lead to irrational decisions and non-optimal use of resources. A study with 53 European hospital managers showed that organizational aspects of new treatments were considered among the most relevant information for decision-making, with safety, clinical and economic aspects [8].

This finding has led to a need for effective decision support tools to enable a systemic approach to decision-making issues [9]. According to a special task force, to identify, collect and structure information required by those making judgments to support the deliberative process, particularly in the field of health, using a multiple criterion decision analysis (MCDA) is a correct route to go down [10]. Developed in the 1970s and widely used in non-medical fields such as agriculture, energy and marketing [11–13], MCDA approaches have spread to the health field since the 2000s [14]. MCDA is especially appropriate to questions requiring a range of relevant but often heterogeneous criteria to be taken on board. It facilitates strategic and operational decision-making in fuzzy or unstable settings, identifying the decision-making problem and shedding useful light on complex issues by implementing a number of alternatives. It also has the advantage of taking account of all relevant aspects (systemic approach), in a multidisciplinary setting, stimulating exchange in a structured decision-making process and enhancing the transparency and rationality of the decisions. To cite different examples of the

value of using this methodology: the Latin American Federation of the Pharmaceutical Industry (FIFARMA) stated that MCDA should strongly be considered as a tool to support Health Technology Assessment (HTA) [15]; and MCDA has been recognized as a relevant approach by the independent expert advisory committee of the Canadian Health Ministry in the context of drug reimbursement decision-making [16]. Recently, the French National Authority for Health (HAS) published, in December 2020, a methodology guide about organizational impact map for health technology assessment based on an MCDA approach [17]. In the MCDA method, the revelation of preferences can be obtained via the discrete choice approach (DCE) or via the Potentially all pairwise rankings of all possible alternatives (PAPRIKA) approach. The DCE method compares scenarios composed of four to six criteria on average [18], whereas the PAPRIKA method compares pairs of scenarios by evaluating two criteria at a time [19]. In the DCE method, the comparison of two alternatives by evaluating more than two criteria can lead to a bias if the user concentrates on one part of the criteria. The analysis conducted is then not exhaustive and can lead to uncertain results [18]. Conversely, the PAPRIKA method allows to simplify the choice of the users. Implicit pairs are automatically eliminated, which reduces the number of pairs to be compared and avoids inconsistencies in classification and the risk of redundancy [18]. This PAPRIKA method is integrated with an online software 1000minds® [20] which has the advantage of generating a random order of comparisons thus reducing the risk of bias related to the order. Moreover, the system does not involve any additional calculation and generates the global (mean or median) and individual weights for each criterion.

In the case before us, the MCDA approach has already been used in the evaluation of the organizational impacts of a therapeutic innovation, such as in acute pain management in an emergency department [9]. Despite some limitations of modeling, this demonstration provided conclusive results. To our knowledge, only one study has used MCDA in the setting of hemophilia. Gourzoulidis et al. (2021) determined' the value of prophylaxis versus on-demand treatment in hemophilia A and emicizumab versus replacement therapy in the Greek healthcare setting [21].

Hemophilia management requires complex structured organization comprising specialized multidisciplinary teams with numerous interfaces with the various actors involved in patient management. Many parameters are relevant to treatment decision-making: type of hemophilia, severity, comorbidities, age, and the patient's personal experience [22]. Introducing several innovations in this context is liable to induce changes in care management organization.

This context appears appropriate to develop the MCDA methodology for the assessment of the organizational impacts of a therapeutic innovation. In our view, this seems relevant because the more complex the system, the greater the uncertainty, especially in biopsychosocial models [23]. Moreover, working to reduce uncertainty will also improve the accompaniment of innovation-induced change, and make greater use of the benefits for improved health.

Thus, we conducted a pilot study in a single hemophilia care center to evaluate the organizational impact of innovation, using MCDA.

## Materials and methods

The Scientific and Ethical Committee of Hospices Civils de Lyon IRB n° 00013204 (Lyon, France) examined the protocol registered under file number 21_322 and waived the requirement for ethics approval. According to French regulations and General Data Protection Regulation, all participants are informed and given the right to object. Since no medical or personal data was collected, informed oral consent was collected during the first meeting and documented in the meeting report.

The study methodology conformed to that of Lvovschi et al. [9]. The good practice guidelines of the International Society for Pharmacoeconomics and Outcomes Research (ISPOR) for the implementation of MCDA by health professionals were followed. Quantitative MCDA was used, as its methodology is well described, it is mainly used in the health field, and it has the advantage of providing aggregate scores that facilitate decision-making.

Seven steps, following the ISPOR quantitative MCDA approach, were followed: (i) defining the decision problem, (ii) selecting and structuring criteria, (iii) weighting criteria, (iv) measuring performance of alternatives, (v) generating a performance score, (vi) calculating aggregate scores, (vii) and deliberation [24, 25].

## Defining the decision problem

The main study objective was to assess the impact of an innovation on the organization of care management of hemophilia patients. Secondary objectives were to define the main criteria underlying this organization and to collect the opinions of those involved regarding the interest of the MCDA approach. The analysis was performed from the point of view of health professionals specializing in hemophilia. Through this procedure organizational impact scores were produced to determine the impact of an innovation on the organization of care: weak impact, requiring minor changes, or strong impact requiring considerable changes.

A multidisciplinary expert group was set up. Inclusion criteria comprised (i) involvement in patient management in the center, (ii) familiarity with care organization, and (iii) being a medical or non-medical health professional. Individuals recruited were identified following an initial study on the care pathway of patient with hemophilia [26]. Among the 11 health professionals previously identified, only the biologist declined to participate. This led to 10 inclusions: 3 physicians, 3 nurses, 2 pharmacists, 1 physiotherapist and 1 psychologist. The study was conducted by a research team with experience in MCDA.

The study took place in the hemophilia treatment center of Chambéry in the Auvergne-Rhône-Alpes area of France, from January to June 2021, with 2 face-to-face meetings of 2 hours each and a remote individual work session (Fig 1).

In the first meeting, the MCDA approach and its interest for assessing the organizational impact of innovations were presented to the group, which then chose to score the organizational impact of the 6 therapeutic innovations shown in Table 1 [2, 3, 5, 27, 28].

## Selecting and structuring criteria

Organizational criteria were selected by the group from a list based on a review of the literature, and grouped in 5 domains: task, surrounding, peoples, structure and technology [29]. Each expert consulted the criteria for each domain to select those that seemed relevant with respect to the predetermined issue. If the proposed criteria did not seem adequate, they could add one in a "free text" field. They were asked to select at least 1 per domain and not more than 8 criteria per domain.

Criteria were validated by consensus after this individual selection stage in the first meeting. The performance levels for each criterion was then established by consensus.

## Weighting criteria

Weighting used the PAPRIKA method (*Potentially all pairwise rankings of all possible alternatives*), a method to reveal preferences integrated in an on-line software (1000minds Ltd, Dunedin, New Zealand) [19]. The principle is based on pairwise comparison between scenarios, assessing two criteria at a time and assuming all others to be equal. Before weighting a criterion, particular attention should be paid to performance level: low, intermediate, high. For the

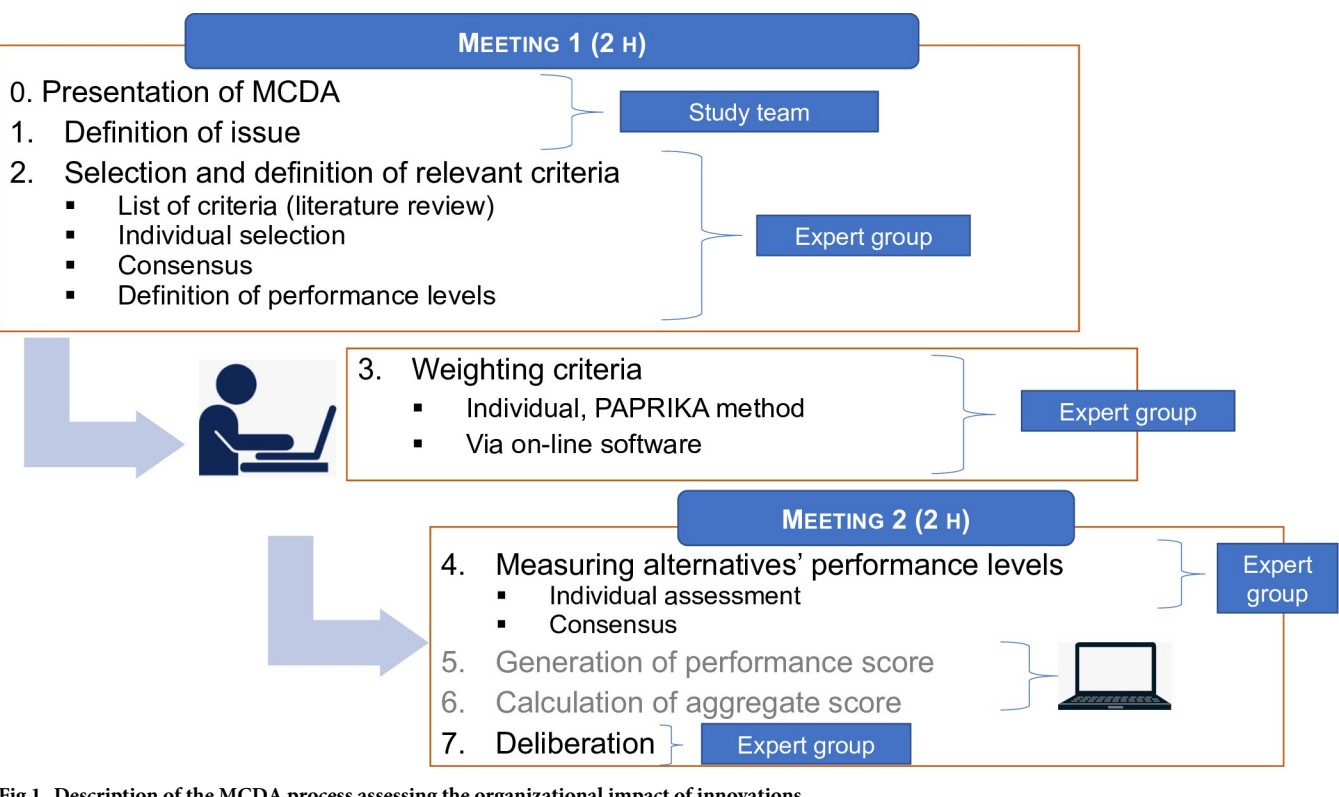

**Fig 1. Description of the MCDA process assessing the organizational impact of innovations.**

study, each expert chose the situation they would prefer, answering the following question: "In the context of a novel hemophilia treatment, which situation would you prefer (in terms of

**Table 1. Alternatives selected by the expert group.**

| THERAPY | TYPE OF THERAPY | INNOVATIVE CHARACTERISTIC | ACCESS TO THERAPY |
|---|---|---|---|
| BIVV001 | Replacement (recombinant factor VIII) | Fusion protein with XTEN polypeptides to extend half-life | Clinical trial (phase III) (hemophilia A) |
| OCTA101 | Replacement (recombinant factor VIII) | Subcutaneous anti-hemophilia factor | Clinical trial (phase I/II) (hemophilia A) |
| emicizumab | Non-replacement (bispecific antibodies mimicking factor VIII action) | Action mechanism: enhancing coagulation. Long half-life. Subcutaneous route | Marketed (hemophilia A including patients with inhibitors) |
| concizumab | Non-replacement (anti-TFPI monoclonal antibodies) | Action mechanism: rebalancing coagulation. Long half-life. Subcutaneous route | Clinical trial (phase III) (hemophilia A and B including patients with inhibitors |
| fitusiran | Non-replacement (RNA interference against antithrombin) | Action mechanism: rebalancing coagulation. Long half-life. Subcutaneous route | Clinical trial (phase III) (hemophilia A and B including patients with inhibitors) |
| volactocogene roxaparvovec giroctocogene fitelparvovec fidanacogene elaparvovec etranacogene dezaparvovec | Gene therapy | Healthy gene introduced in a single administration | Clinical trial (phase III) (hemophilia A and B including patients with inhibitors) |

impact on the organization of care)?" The upper limit thus corresponded to the lowest-impact situation and the highest weighting.

An explanatory notice was drawn up to optimize use of the tool. The experts provided their individual preference via the software. The mean weighting was used to calculate the aggregate score.

## Measuring performance of alternatives

The second meeting reviewed the main characteristics of the innovations according to the current state of knowledge. The performance level of the alternatives was then assessed by consensus after a step of individual assessment.

## Scoring alternatives and calculating aggregate scores

The performance scores and aggregate score of each innovation were calculated automatically by the PAPRIKA software after entering the performance levels validated by the expert group.

The method for aggregating weightings and performance levels most often used in the health field is the weighted sum method:

$$S(a) = \sum_{i=1}^{n} w_i s_i(a)$$

where S(a) is the aggregate score, w the relative weight of the criterion, and $s_i(a)$ the performance score for alternative A for criterion $i$.

## Results

### Choice and description of criteria

Eight criteria were identified as priorities in the first meeting: 2 in the "task" domain, 1 in "surrounding", 1 in "peoples", 1 in "structure" and 2 in "technology"." The "structure" criterion (innovation matching network objectives) was eliminated in the assessment meeting; at care-unit level, it is rather a precondition for introducing an innovation rather than a criterion that varies with department organization depending on the innovation in question. Performance levels per criterion ranged between 3 and 4; Table 2 shows criteria and performance levels.

### Weightings of criteria

The experts made between 46 and 84 explicit comparisons in a median 14 minutes (range, 9–120 min). Those who took longest (55 and 120 minutes) did not complete the analysis, due to interruption: 1 nurse (who began again), and 1 physiotherapist; the physiotherapist's partial analysis was not taken into account in weighting the criteria.

The 3 criteria with the strongest weightings were "acceptability for patient/care-giver/association" (weight: 31.4%), "number of emergency or unscheduled consultations" (weight: 20.3%) and "complexity of the innovation and informed choice" (weight: 16.6%). Fig 2 shows criterion weightings.

### Assessment of organizational impact of innovations

Alternatives were assessed by 4 of the experts: 1 physician, 2 nurses and 1 pharmacist. Responses varied according to profession and individual factors of understanding the criteria and knowledge of the innovation. After compiling individual results, discrepancies were discussed and consensus was reached for all innovations. The "number of emergency or

**Table 2. Criteria for evaluation of the organizational impact.**

| Field | Criteria | Items | Item definitions |
|---|---|---|---|
| Task | Number of emergency or unscheduled consultations | Strong increase | Increase in number of emergency and unscheduled consultations |
| | | Increase | Increase in number of unscheduled consultations |
| | | No change | No change |
| | | Decrease | Decrease in number of emergency consultations |
| | | Strong decrease | Decrease in number of emergency and unscheduled consultations |
| | Change in patient recruitment | Complexification | Need for additional decision criteria |
| | | No change | No change |
| | | Simplification | Innovation meeting non-covered therapeutic need |
| Surrounding | Number of pathway or interface actors | Increase | New actors along pathway |
| | | No change | No change |
| | | Decrease | Fewer actors along pathway |
| Peoples | Medical and non-medical personnel training | Considerable | Very complex theoretical and practical training requiring development of several supports; long training (several days) |
| | | Classical or standard | Complex theoretical and practical training requiring development of a single support accessible to all types of health professional; duration <1 day. |
| | | Lightened | Simple theoretical and practical training requiring no new support; duration negligible (<1h) |
| | | None | No training required |
| Structure | Acceptability for patient/care-giver/association | None | No contribution in terms of efficacy, tolerance, practicality, adherence |
| | | Weak | Improves 1 criterion of efficacy, tolerance, practicality, adherence |
| | | Moderate | Improves 1 or 2 criteria of efficacy, tolerance, practicality, adherence |
| | | Strong | Improves several criteria of efficacy, tolerance, practicality, adherence |
| Technology | Supply (product, associated devices) | Very complex | Several ordering procedures for products and devices, different supply procedures and supply times |
| | | Complex | One additional step, easy to include in current procedures |
| | | Simple | No change |
| | Complexity of innovation and informed choice | Very complex | Action mechanism hard to understand even for professionals; difficult to popularize |
| | | Complex | Action mechanism more difficult to popularize |
| | | Simple | Action mechanism easy to explain |

unscheduled consultations" criterion had the same score for all innovations (13.3 points) except for gene therapy (18.7 points). This was accounted for by the small number of emergency or non-scheduled consultations in everyday practice, leaving little room for improvement on this criterion. Treatments were presumed to be effective in the light of the state of knowledge at the time of the study. Thus, no deterioration in this criterion was expected. Responses to the "supply" criterion revealed a need for better definition, taking account of the laboratory supplying the treatment. For treatments still at the stage of clinical trials, supply modalities were difficult to determine. Obviously, the experts assessed this criterion taking account of the laboratory's expertise: hindsight on the knowledge of the pathology and patients' needs in terms of associated devices and the ability to set up emergency supply routes. One innovation could not be assessed on all criteria, as the experts lacked knowledge of the treatment. The 2 innovations with the highest aggregate scores (i.e., low impact on care organization) were subcutaneous recombinant anti-hemophilia factor (87/100) and very long action recombinant anti-hemophilia factor (88.8/100). Low scores (i.e., strong impact) were attributed to gene therapy (27.3/100) and anti-antithrombin RNA (21.9/68.3); their acceptability was deemed low and the innovation was considered difficult to explain in securing informed patient choice. Table 3 shows aggregate scores per innovation.

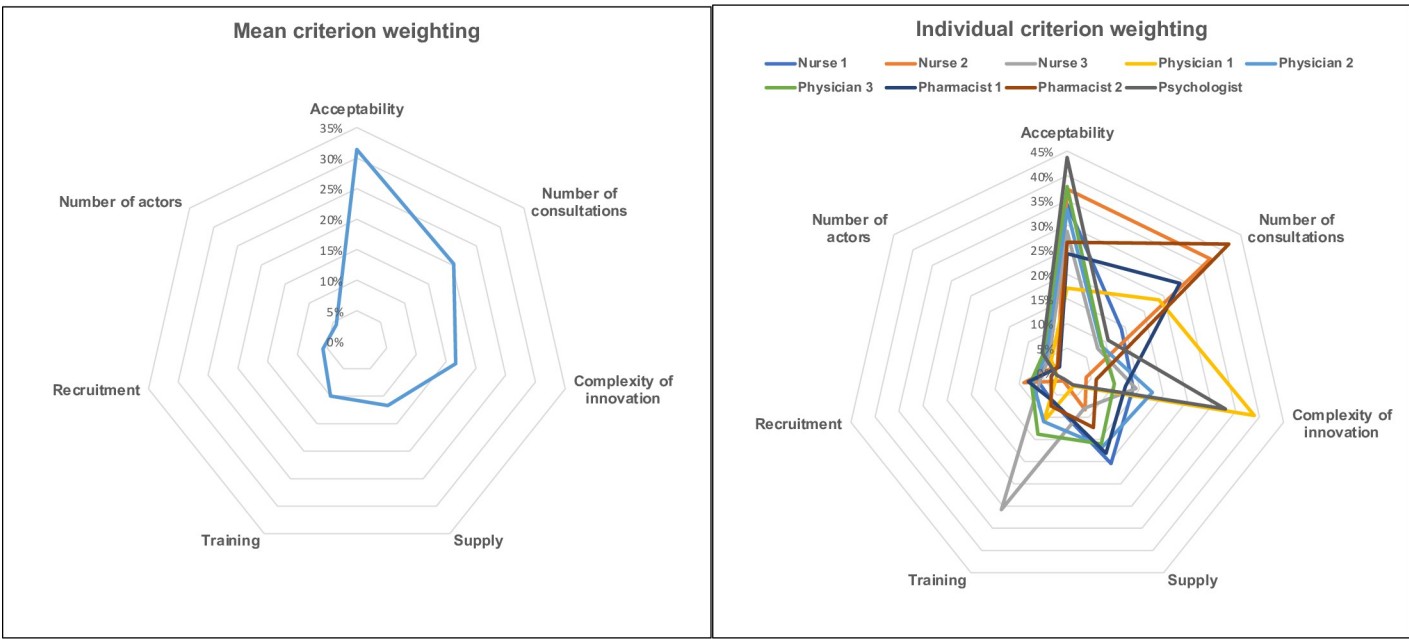

**Fig 2. Mean and individual weighting per criterion.**

## Discussion

This study was conducted in the field of hemophilia care. And as it took place in France, some local conditions and standards of care must be taken into account. For example, while France has implemented all 10 principles of hemophilia care (while other European countries have not), it is recognized that various aspects of the French hemophilia patient's care need to be improved (aging, psychosocial support, etc.) [27, 30, 31]. All the more so with regard to the therapeutic innovations that are coming.

Therefore, this study shows that the use of a structured analysis method helps to formalize the essential organizational criteria in the management of a pathology and to become aware of the possible impacts in the event of a breakthrough innovation in therapies. Thus, therapeutic innovations can induce organizational innovations, and recognizing this early and precisely can make it possible to prepare adequate support for change for care teams.

Then, the MCDA method, used in this study, allowed an expert group to attribute organizational impact scores for each study innovation. The method assessed the organizational impact

**Table 3. Organizational impact scores per innovation.**

|  | BIVV001 | OCTA101 | emicizumab | concizumab | fitusiran | gene therapy |
|---|---|---|---|---|---|---|
| Acceptability | 31.4 | 31.4 | 31.4 | 19.3 | 8.6 | 8.6 |
| Number of consultations | 13.3 | 13.3 | 13.3 | 13.3 | 13.3 | 18.7 |
| Complexity | 16.6 | 16.6 | 8.1 | 8.1 | 0.0 | 0.0 |
| Supply | 11.7 | 11.7 | 6.1 | 11.7 | 0.0* | 0.0 |
| Training | 7.7 | 7.7 | 7.7 | 7.7 | 0.0* | 0.0 |
| Recruitment | 3.9 | 5.7 | 0.0 | 0.0 | 0.0* | 0.0 |
| Number of actors | 2.4 | 2.4 | 0.0 | 0.0 | 0.0* | 0.0 |
| Aggregate score | 87 | 88.8 | 66.6 | 60.1 | 21.9 | 27.3 |

* In the absence of sufficient data in the current state of knowledge

of innovations and identified useful points for reflection for a subsequent more qualitative analysis of the nature of such organizational changes. An innovation that is complex to explain may require setting up new explanatory tools, impacting care-unit organization. In contrast, innovations consisting in replacement showed the least organizational impact, although this is to be read in the context of the study situation and cannot be directly extrapolated to other hemophilia centers. Although hemophilia management is highly structured and coordinated, there may be organizational differences between centers according to their geographic situation and history. It would be useful to repeat the present study in other centers, to analyze variations in choice of relevant organizational criteria and possible differences in impact according to care organization and geographical region. This could complement pathway analysis, with optimization according to organizational specificities, and also a greater consideration of complementary actors, like city caregivers. For example, in France, since June 15, 2021 (after this study), the pharmaceutical product emicizumab has had a dual dispensing circuit: hospital pharmacy and city pharmacy. This is the first hemophilia treatment available to patients in private practices. This dispensing is accompanied by additional risk-reduction measures, like a new organizational chart [32], an optimized information flow and a full training on the disease and treatments.

A better understanding of the nature and strength of the organizational impact of a therapeutic innovation should allow for a better match between therapeutic choice and individual's health determinants (biological characteristics, family environment, local health system, cultural context, etc.), to understand the most suitable patient support, and to ensure an optimal healthcare pathway, reducing the risk of dropouts.

From the point of view of the experts, the acceptability for patients/caregiver/association, the number of emergency and unscheduled consultations and the complexity of the innovation were the main criteria determining organizational impact. Gene therapy and anti-antithrombin RNA were the innovations with the greatest organizational impact, notably due to poor acceptability and high complexity hindering shared informed decision-making. Acceptability is a multifactorial criterion that may be related more to the patient and their experience and life than to the innovation itself. Thus, for example, not taking into account a good match between individual capacity for acceptability and the complexity of the innovation, which may lead to difficulties in understanding its usefulness or usability, could lead to a risk of poor therapeutic adherence. In this case, having already planned a specific work to increase the patient's level of health literacy could be a key success factor. Poor therapeutic adherence could also result from supply difficulties inherent to a therapeutic innovation if these were not clearly envisaged beforehand.

At patient level, individual variation in organizational impact score can be expected due to the strong weighting of this criterion. In a qualitative study in 20 hemophilia patients, 40% were very favorable and 35% favorable toward gene therapy, after information delivered ahead of the interview [33]. This study underlined similarities in the organizational impact from gene therapy and anti-antithrombin RNA related to their mechanism of action. It was estimated as very complex by this expert group. The organizational impact of these two therapeutic innovations differed in terms of the number of emergency or unscheduled consultations. This criterion was considered favorable only for gene therapy because of its effectiveness in treating hemophilia. New tools or process need to be implemented in order to bring appropriate information at patient for shared-decision. This need is not specific to this team since other studies stressed the need to develop means of training or communication to facilitate shared decision-making [34–37], in agreement with the present expert group's opinion regarding the importance of information for shared decision-making in hemophilia management. In this sense, it has already been demonstrated elsewhere that gene therapy brings a different therapeutic value chain than

traditional pharma, and involves new capabilities and requirements [38]. It calls for a rethink of current collaboration: additional roles of a hemophilia treatment centers could be implemented, and 'hub-and-spoke' models are thought out (prescription and management in national hubs; monitoring in spokes centers) [39]. This makes it crucial to educate patient and health care personnel, essential to define new quality and safety criteria [38].

The present study contributes to the early phase of assessment of MCDA in organizational impact in the health field. The MCDA method aims to facilitate the identification of the optimal solution [40]. It is particularly appropriate for problems that require taking into account a set of relevant and often heterogeneous criteria. It facilitates strategic or operational decision-making in an imprecise or unstable environment, objectifies a decision-making problem, and analyzes complex problems by implementing several possible alternatives. This method has several advantages. It allows (i) to take into account all relevant aspects (systemic approach), (ii) to be part of a multidisciplinary approach, (iii) to stimulate exchanges within the framework of a structured decision-making process, and (iv) to improve the transparent and reasoned character of decision-making. However, the MCDA method has several drawbacks [24, 41]. First, sufficient material and human resources (time, expertise, commitment) must be available and planned. In addition, training and familiarization of professionals are prerequisites for the efficient implementation of these methods. On the other hand, the limitations of multi-criteria methods are mainly related to reliability (accurate and complete data), variability and level of evidence of the input data, and structural uncertainty, such as disagreement on the weighting method. In this regard, several limitations are to be borne in mind.

The composition of the group immediately follows the starting point of any multi-criteria method, the definition of the problem may present biases in this study. It has an impact on the choice of alternatives, criteria and performance levels that structure the problem. Moreover, the group, composed of individuals from different backgrounds, is at the origin of the weighting of the criteria and their performance levels. Variations are expected; the aggregation of individual results with or without (average value) group consensus allows participants to express their point of view. Its composition is a crucial element of the multi-criteria method [10]. It is a source of heterogeneity which must be controlled via sensitivity analysis (study of subgroups, exploration of uncertainties and disagreements with other groups) [24].

A single workshop with participants may not be sufficient to ensure a representative evaluation, and multiple workshops or surveys may be necessary [24]. In this exploratory work, two workshops were held and the individual preference revelation step was conducted remotely.

Regarding the number of experts in the group, two other studies were performed with the same method, comprising 8 and 7 experts respectively [9, 40]. However, studies need repeating to test the method in larger groups so as to resolve issues of heterogeneity and assess varied alternatives.

No patients were included in the group, although patients play a growing role in decisions affecting their health. Including them in the decision-making process is coherent with the development of pathway-based approaches and of interactions with professionals to achieve transverse dialogue. The present study will be followed up by a qualitative study in patients regarding their perception of the impact on their care organization [42]. Finally, training participants is essential, and in the present study the experts received no specific training, although the MCDA method was presented at the outset of the study.

Regarding the PAPRIKA weighting method, several authors reported that users could find it difficult to decide about certain scenarios. They find it difficult to choose between two alternatives while setting aside important criteria not involved in the proposed scenarios. The present expert group did not raise this issue. Goetghebeur et al. also reported that some users found it hard to express their own point of view rather than projecting into an institutional or

social perspective. The perspective in the present study was that of the treatment center. PAPRIKA is an ordinal method, more robust than those used in grading scales for decision aid such as EVIDEM (Evidence and Value: Impact on DEcisionMaking) or Matrix4Value® [43–45]. It avoids several biases inherent to grading scales and makes comparison of weightings more reliable. The problem of individual perception of the scale is circumvented. And comparing two alternatives strictly in terms of two criteria prevents the user concentrating on only one part of the criteria.

The individual preferences expressed by the experts were seen as homogeneous by the group as a whole. However, individual weightings were closer for the criterion with the greatest weight (acceptability). Other authors likewise reported comparable weightings between assessors for the criteria judged more important; in the present study, individual weightings for criteria judged to be of intermediate importance (training and supply) were more divergent. This was also found by Goetghebeur et al. and Martelli et al.. According to Goetghebeur et al., divergences in weighting corresponded to varying individual perceptions and representations, and were not an obstacle; rather, they contribute to the exchanges generating consensus. These methods are interesting as they reveal disagreements between participants.

The same experts determined the weightings of the criteria and assessed the alternatives on these criteria. Other authors proceeded in the same way; however, Sampietro-Colom. et al. suggested that this could introduce bias. The Matrix4value® method uses two groups of experts; but this requires having sufficient human resources.

A question arises concerning maintaining non-discriminating criteria in assessing alternatives. In the present study, number of consultations was non-discriminating between the innovations with the exception of gene therapy. However, these criteria still have an impact. Increasing the number of performance levels for these criteria could confirm or discount their non-discriminating character. Moreover, in case of doubt regarding a difference in performance between several alternatives for a given criterion, the same score was attributed, and the criterion became non-discriminating, without, however, being canceled. Opinions differ on this point. Martelli et al. and Sampietro-Colom et al. attributed a score of zero in case of doubt, whereas Goetghebeur et al. attributed a low score in the EVIDEM model.

No sensitivity analysis was performed in the present study. Our MCDA model had several sources of uncertainty, affecting all components [24, 41]. Structural uncertainty was presumed to be low, as the weighting method and criterion value tree were established consensually. Uncertainty mainly came from using expert opinions as entry data. If a performance measure had been used, the choice between introducing an uncertainty criterion or an uncertainty analysis could have been discussed [24]. Oliveira et al. reported that different approaches to uncertainty analysis are used in MCDA to assess health technologies [46]. Moreover, the present low level of structural uncertainty could be confirmed by scenario analysis [24]. Finally, heterogeneity of preference could be studied by calling upon other expert groups [24].

In the absence of any prior assessment of the impact of the selected innovations on the organization of hemophilia care, no comparison could be made between a previous assessment process and the present method, as proposed by Martelli et al. and Goetghebeur et al. [43, 44]. Simulations could be set up to this end.

## Conclusion

The interest of studying organizational impact is now recognized and including it in health technology assessment is a goal for the coming years. The idea of cost-effectiveness is now well understood by decision-makers. However, what choice should be made about an innovation of clear clinical efficacy but requiring changes in pathway organization and relations between

actors? The distinction between medico-economic and organizational evaluation needs to be retained; but the study of organizational impact can enrich medico-economic evaluation. The question of level of evidence arises for the organizational dimension. Moreover, taking organizational impact into account complexifies the evaluation of innovations. Methods must be chosen that respect a balance between simplicity of implementation and methodological rigor. It is also important to decide at what point in the cycle of an innovation organizational impact can most usefully be assessed.

## Acknowledgments

We acknowledge the members of the hemophilia treatment center from Chambery, especially Isabelle, Laurence, Odile, V. Favre, B. Steng, Dr Sylvestre, Dr Jailler and Dr Meyer for their participation in this study. We would like to thank Mr Iain McGill for his help in manuscript preparation.

## Author Contributions

**Conceptualization:** Karen Beny, Amélie Dubromel, Benjamin du Sartz de Vigneulles, Claude Dussart.

**Data curation:** Karen Beny, Amélie Dubromel, Benjamin du Sartz de Vigneulles.

**Formal analysis:** Karen Beny, Amélie Dubromel, Benjamin du Sartz de Vigneulles.

**Investigation:** Karen Beny, Amélie Dubromel, Benjamin du Sartz de Vigneulles, Valérie Gay.

**Methodology:** Karen Beny, Amélie Dubromel, Benjamin du Sartz de Vigneulles, Claude Dussart.

**Project administration:** Benjamin du Sartz de Vigneulles.

**Resources:** Valérie Gay, Claude Dussart.

**Supervision:** Claude Dussart.

**Validation:** Claude Negrier.

**Writing – original draft:** Karen Beny, Amélie Dubromel, Benjamin du Sartz de Vigneulles.

**Writing – review & editing:** Valérie Gay, Florence Carrouel, Claude Negrier, Claude Dussart.

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
