## [Decision Letter · Decision Letter 0]

21 Jun 2022

PONE-D-21-37437Multiple criteria decision analysis for therapeutic innovations in a hemophilia care center: the organizational impact of innovation in hemophilia care management.PLOS ONE

Dear Dr. Beny,

Thank you for submitting your manuscript to PLOS ONE. After careful consideration, we feel that it has merit but does not fully meet PLOS ONE’s publication criteria as it currently stands. Therefore, we invite you to submit a revised version of the manuscript that addresses the points raised during the review process.

We look forward to receiving your revised manuscript.

Kind regards,

Wolfgang Miesbach, MD

Academic Editor

PLOS ONE

Journal Requirements:

Reviewers' comments:

Reviewer's Responses to Questions

**Comments to the Author**

1. Is the manuscript technically sound, and do the data support the conclusions?

Reviewer #1: Yes

Reviewer #2: Yes

2. Has the statistical analysis been performed appropriately and rigorously? 

Reviewer #1: Yes

Reviewer #2: Yes

3. Have the authors made all data underlying the findings in their manuscript fully available?

Reviewer #1: Yes

Reviewer #2: Yes

4. Is the manuscript presented in an intelligible fashion and written in standard English?

Reviewer #1: Yes

Reviewer #2: Yes

5. Review Comments to the Author

Reviewer #1: This is a potential interesting manuscript on the potential organizational impact of new therapeutic options for the treatment of hemophilia. The authors used a decision analysis based on several criteria.

Generally of interest, several points should be clarified.

The advantages and disadvantages of the MCDA method should be discussed, as well as some limitations.

What possible consequences can be drawn from the results of this study? Some haemophilia experts should comment.

Some of the requirements for gene therapy (described in the hub-and-spoke model) should be added.

Since this study was conducted in France, some of the local conditions and standard of care should be considered.

What are the organizational implications of gene therapy for hemophilia and anti-antithrombin RNA? Are there any similarities or dissimilarities?

Please correct some typos, e.g., futisuran, etc.

Reviewer #2: Thank you for your submission which makes an important contribution to our understanding of the changing treatment landscape.

There are however a couple of points that I feel need clarification and or amendment.

Line 74

You state that “to our knowledge, MCDA has never been applied to haemophilia innovations”.

This maybe correct but there have been a number of discrete choice experiments (DCE), including that of Woollacott et al (2022) that have examined professional perspectives on new haemophilia treatments. Is there a significant enough difference between an MCDA and a DCE approaches that favours MCDA in this context? I think the introduction would benefit from a short discussion on this and why you chose MCDA over DCE.

Line 85

You state that this is a pilot study. This should be reflected in both the title and in the abstract.

Line 112ff

You state how many individuals were recruited and their composition but do not mention how many were approached, declined to participate and what the composition of this group was.

Also you do not discuss whether there were any possible biases in the group. Again, some discussion of this would benefit the paper.

6. PLOS authors have the option to publish the peer review history of their article (what does this mean?). If published, this will include your full peer review and any attached files.

Reviewer #1: No

Reviewer #2: No

---

## [Author Response · Author response to Decision Letter 0]

8 Jul 2022

Dear reviewers,

Thank you for careful and thorough reading of this manuscript and for the thoughtful comments and constructive suggestions, which help to improve the quality of this manuscript. Our response follows (the reviewer’s comments and our responses).

Reviewer #1: This is a potential interesting manuscript on the potential organizational impact of new therapeutic options for the treatment of hemophilia. The authors used a decision analysis based on several criteria.

Generally of interest, several points should be clarified.

#1 The advantages and disadvantages of the MCDA method should be discussed, as well as some limitations.

We added line 278 : 

« The MCDA method [43] aims to facilitate the identification of the optimal solution. It is particularly appropriate for problems that require taking into account a set of relevant and often heterogeneous criteria. It facilitates strategic or operational decision-making in an imprecise or unstable environment, objectifies a decision-making problem, and analyzes complex problems by implementing several possible alternatives. This method has several advantages. It allows (i) to take into account all relevant aspects (systemic approach), (ii) to be part of a multidisciplinary approach, (iii) to stimulate exchanges within the framework of a structured decision-making process, and (iv) to improve the transparent and reasoned character of decision-making. However, the MCDA method has several drawbacks. First, sufficient material and human resources (time, expertise, commitment) must be available and planned. In addition, training and familiarization of professionals are prerequisites for the efficient implementation of these methods. On the other hand, the limitations of multi-criteria methods are mainly related to reliability (accurate and complete data), variability and level of evidence of the input data, and structural uncertainty, such as disagreement on the weighting method. In this regard, several limitations are to be borne in mind.”

#2 What possible consequences can be drawn from the results of this study? Some haemophilia experts should comment.

We added some comments in the discussion section. See lines 221-225, 238-247 and 253-258. 

#3 Some of the requirements for gene therapy (described in the hub-and-spoke model) should be added.

We added line 271-277 : 

“In this sense, it has already been demonstrated elsewhere that gene therapy brings a different therapeutic value chain than traditional pharma, and involves new capabilities and requirements [39]. It calls for a rethink of current collaboration: additional roles of a hemophilia treatment centers could be implemented, and ‘hub-and-spoke’ models are thought out [40] (prescription and management in national hubs; monitoring in spokes centers) [41]. This makes it crucial to educate patient and health care personnel, essential to define new quality and safety criteria [42].”

#4 Since this study was conducted in France, some of the local conditions and standard of care should be considered.

We added some information in the discussion section. See lines 216-225, 238-247 and 253-258. 

#5 What are the organizational implications of gene therapy for hemophilia and anti-antithrombin RNA? Are there any similarities or dissimilarities?

We added line 262: “This study underlined similarities in the organizational impact from gene therapy and anti-antithrombin RNA related to their mechanism of action. It was estimated as very complex by this expert group. The organizational impact of these two therapeutic innovations differed in terms of the number of emergency or unscheduled consultations. This criterion was considered favorable only for gene therapy because of its effectiveness in treating hemophilia. New tools or process need to be implemented in order to bring appropriate information at patient for shared-decision. This need is not specific to this team since other studies stressed the need to develop means of training or communication to facilitate shared decision-making [36-38], in agreement with the present expert group’s opinion regarding the importance of information for shared decision-making in hemophilia management.”

#6 Please correct some typos, e.g., futisuran, etc.

Typos have been corrected.  

Reviewer #2: Thank you for your submission which makes an important contribution to our understanding of the changing treatment landscape.

There are however a couple of points that I feel need clarification and or amendment.

#6 Line 74 You state that “to our knowledge, MCDA has never been applied to haemophilia innovations”. 

Indeed, when we submitted the article, no MCDA study analyzing hemophilia was published to our knowledge. However, we conducted a new literature search that revealed a study published in December 2021 (after our submission). Therefore, we have replaced the sentence line 84 with: "To our knowledge, only one study has used MCDA in the setting of hemophilia. Gourzoulidis et al (2021) determined' the value of prophylaxis versus on-demand treatment in hemophilia A and emicizumab versus replacement therapy in the Greek healthcare setting.”

This maybe correct but there have been a number of discrete choice experiments (DCE), including that of Woollacott et al (2022) that have examined professional perspectives on new haemophilia treatments. Is there a significant enough difference between an MCDA and a DCE approaches that favours MCDA in this context? I think the introduction would benefit from a short discussion on this and why you chose MCDA over DCE.

We added line 69 : « In the MCDA method, the revelation of preferences can be obtained via the discrete choice approach (DCE) or via the Potentially all pairwise rankings of all possible alternatives (PAPRIKA) approach. The DCE method compares scenarios composed of four to six criteria on average [18], whereas the PAPRIKA method compares pairs of scenarios by evaluating two criteria at a time [19]. In the DCE method, the comparison of two alternatives by evaluating more than two criteria can lead to a bias if the user concentrates on one part of the criteria. The analysis conducted is then not exhaustive and can lead to uncertain results [20]. Conversely, the PAPRIKA method allows to simplify the choice of the users. Implicit pairs are automatically eliminated, which reduces the number of pairs to be compared and avoids inconsistencies in classification and the risk of redundancy [20]. This PAPRIKA method is integrated with an online software 1000minds® [21] which has the advantage of generating a random order of comparisons thus reducing the risk of bias related to the order. Moreover, the system does not involve any additional calculation and generates the global (mean or median) and individual weights for each criterion [22].

#7 Line 85 You state that this is a pilot study. This should be reflected in both the title and in the abstract.

We completed the title and the abstract as recommended.

#8 Line 112 You state how many individuals were recruited and their composition but do not mention how many were approached, declined to participate and what the composition of this group was.

We added line 127: “Individuals recruited were identified following an initial study on the care pathway of patient with hemophilia [27]. Among the 11 health professionals previously identified, only the biologist declined to participate.”

#9 Also you do not discuss whether there were any possible biases in the group. Again, some discussion of this would benefit the paper.

We added line 293: « The composition of the group immediately follows the starting point of any multi-criteria method, the definition of the problem may present biases in this study. It has an impact on the choice of alternatives, criteria and performance levels that structure the problem. Moreover, the group, composed of individuals from different backgrounds, is at the origin of the weighting of the criteria and their performance levels. Variations are expected; the aggregation of individual results with or without (average value) group consensus allows participants to express their point of view. Its composition is a crucial element of the multi-criteria method [43]. It is a source of heterogeneity which must be controlled via sensitivity analysis (study of subgroups, exploration of uncertainties and disagreements with other groups) [44].

A single workshop with participants may not be sufficient to ensure a representative evaluation, and multiple workshops or surveys may be necessary. In this exploratory work, two workshops were held and the individual preference revelation step was conducted remotely [44].»

---

## [Editor Report · Decision Letter 1]

16 Aug 2022

Multiple criteria decision analysis for therapeutic innovations in a hemophilia care center: a pilot study of the organizational impact of innovation in hemophilia care management.

PONE-D-21-37437R1

Dear Dr. Beny,

We’re pleased to inform you that your manuscript has been judged scientifically suitable for publication and will be formally accepted for publication once it meets all outstanding technical requirements.

Kind regards,

Wolfgang Miesbach, MD

Academic Editor

PLOS ONE

---

## [Editor Report · Acceptance letter]

1 Sep 2022

PONE-D-21-37437R1 

Multiple criteria decision analysis for therapeutic innovations in a hemophilia care center: a pilot study of the organizational impact of innovation in hemophilia care management. 

Dear Dr. Beny:

I'm pleased to inform you that your manuscript has been deemed suitable for publication in PLOS ONE. Congratulations! Your manuscript is now with our production department. 

Kind regards, 

on behalf of

Dr. Wolfgang Miesbach 

Academic Editor

PLOS ONE